# Integrated Transcriptome and Proteome Analysis Reveals That Cell Wall Activity Affects *Phelipanche aegyptiaca* Parasitism

**DOI:** 10.3390/plants13060869

**Published:** 2024-03-18

**Authors:** Meixiu Chen, Lu Zhang, Zhaoqun Yao, Xiaolei Cao, Qianqian Ma, Siyu Chen, Xuekun Zhang, Sifeng Zhao

**Affiliations:** 1Key Laboratory of Special Fruits and Vegetables Cultivation Physiology and Germplasm Resources Utilization of Xinjiang Production and Construction Corps, Shihezi University, Shihezi 832003, China; 13289938193@163.com (M.C.); 13029617900@163.com (L.Z.); 2Key Laboratory of Oasis Agricultural Pest Management and Plant Protection Resources Utilization, Xinjiang Uygur Autonomous Region, Shihezi University, Shihezi 832003, China; yaozhaoqun@sina.com (Z.Y.); tulanduocxl@sina.com (X.C.); 15379722990@163.com (Q.M.); 13199571099@163.com (S.C.)

**Keywords:** *P. aegyptiaca*, *C. lanatus*, transcriptome, proteome, cell wall, pectinesterase

## Abstract

*Phelipanche aegyptiaca* can infect many crops, causing large agricultural production losses. It is important to study the parasitism mechanism of *P. aegyptiaca* to control its harm. In this experiment, the *P. aegyptiaca* HY13M and TE9M from Tacheng Prefecture and Hami City in Xinjiang, respectively, were used to analyze the parasitical mechanism of *P. aegyptiaca* by means of transcriptome and proteome analyses. The parasitic capacity of TE9M was significantly stronger than that of HY13M in *Citrullus lanatus*. The results showed that the DEGs and DEPs were prominently enriched in the cell wall metabolism pathways, including “cell wall organization or biogenesis”, “cell wall organization”, and “cell wall”. Moreover, the functions of the pectinesterase enzyme gene (*TR138070_c0_g*), which is involved in the cell wall metabolism of *P. aegyptiaca* in its parasitism, were studied by means HIGS. The number and weight of *P. aegyptiaca* were significantly reduced when *TR138070_c0_g1*, which encodes a cell-wall-degrading protease, was silenced, indicating that it positively regulates *P. aegyptiaca* parasitism. Thus, these results suggest that the cell wall metabolism pathway is involved in *P. aegyptiaca* differentiation of the parasitic ability and that the *TR138070_c0_g1* gene plays an important role in *P. aegyptiaca*’s parasitism.

## 1. Introduction

Broomrape (*Phelipanche* spp., Syn. *Orobanche* spp.) is an obligate root holoparasitic plant that causes severe yield losses in crops and is mainly distributed in North Africa, the Mediterranean, Europe, and the Middle East region [1,2]. Broomrape does not possess functional roots and cannot perform effective photosynthesis. Further, it cannot grow without a host, because it obtains all its nutrients from the host [3,4]. Orobanche can infect a wide range of host families, including Solanaceae, Brassicaceae, Cucurbitaceae, Cruciferae, Apiaceae, Fabaceae, and Asteraceae, leading to damage to their hosts [5,6]. In China, muskmelon, an important agricultural product with an annual yield of up to 1.7325 million tons, faces a significant threat from broomrape [7,8]. About 1500–2000 ha of muskmelon are affected by *P. aegyptiaca* each year [9]. It is urgent to develop effective control management for *P. aegyptiaca* to ensure the sustainable development of melon cultivation.

*Orobanche* spp. have evolved a special life strategy that is highly coordinated with their hosts throughout their entire life cycle, rendering herbicides and other management approaches ineffective [10,11]. Various methods have been used to control broomrape, including cultural practices [12], soil solarization, soil fumigation, and chemical control [13,14,15]. However, none of these measures have achieved successful control effects [16] due to broomrape’s biological traits, such as underground parasitism, growth synchronization with the host, uptake of resources via the crop’s vascular system [17], long-lived seed viability [18], and the potential to develop a strong seed bank in the short term [19]. One major strategy for inhibiting broomrape’s ability to penetrate the host is the development of resistant cultivars [17,20,21]. However, the variation in broomrape populations can overcome previously resistant host plants. Understanding the parasitic differences between different *P. aegyptiaca* populations may provide insights into the parasitic mechanism and serve as a basis for the development of prevention methods and control agents.

Increasing evidence shows that the variation in and evolution of parasitic plants lead to varying enzyme activity responses in the parasitic process [22]. Among the alterations, enzyme activity associated with cell walls may determine the virulence of parasitic plants. For example, the pectin methylesterases, polygalacturonases, peroxidases, and chalcone synthases are upregulated in race H compared to race E in *O. cumana* [23]. Cell-wall-degrading enzymes, including pectin methylesterase (PME, EC 3.1.1.11) and polygalacturonase (PG, EC 3.2.1.15), exhibited higher activities in the most virulent race (*O. cumana*) [24]. Similarly, during Orobanchaceae evolution, glycosyl hydrolase and pectate lyase (PL) genes were upregulated in at least two species of Orobanchaceae [22]. Moreover, cell-wall-modifying proteins were revealed to be candidate virulence factors by scanning the genome of *Striga hermonthica*, a root parasitic plant within the Orobanchaceae [25]. Parasitism gene exploration revealed that numerous cell-wall-modifying proteins originated from gene duplications in their ancestor [26]. However, how the parasitic plant exploits cell wall proteases to assist the parasitic process remains to be explored.

*Orobanche* has evolved to adapt to the host by altering cell wall metabolism through different parasitic processes. During the initial penetration stages, the cell walls are degraded by enzymes, such as polygalacturonases (PGUs), pectin methylesterases (PMEs), rhamnogalacturonases, and peroxidases (PRXs), from *Orobanche* spp. [23,27]. These enzymes are one of the reasons for the virulence of *O. cumana* [23,24]. Moreover, the cell wall protease genes associated with penetration and infection are produced in susceptible interactions between *O. cumana* and sunflower [28]. Therefore, proteases relating to cell wall activity are key factors determining the success of parasitism. For example, expansin-like proteins can accelerate cell wall degradation and promote host penetration [29,30]. The β-expansin gene in *Triphysaria versicolor* is upregulated in response to the monocot host *Zea mays*; pectinesterase, polygalacturonase, and cell-wall-modifying enzymes are also highly expressed in *T. versicolor* at the parasite–host interface [31]. Furthermore, in Orobanchaceae plants, cell-wall-modifying enzymes are upregulated during haustorial development following host attachment [22]. Therefore, the cell wall metabolism enzymes of parasitic plants and compositional changes in the host cell wall are necessary for the development of parasitic plants.

In this study, *P. aegyptiaca* populations with different parasitic capacities and *C. lanatus*, *Cucurbita moschata*, and *Lagenaria siceraria* were selected as research materials. Transcriptome and proteome analyses and host-induced gene silencing (HIGS) were employed. This research aimed to (i) evaluate the parasitic ability of different *P. aegyptiaca* populations against *C. lanatus*, *C. moschata*, and *L. siceraria*; (ii) analyze the cell wall activity in different interactions between *P. aegyptiaca* and *C. lanatus*; and (iii) screen out and verify the cell wall metabolism genes in *P. aegyptiaca*. These results provide a theoretical foundation for understanding the mechanism targeting host cell wall activity in the parasitic process of different *P. aegyptiaca* populations.

## 2. Materials and Method

### 2.1. Plant Materials and Growing Conditions

*P. aegyptiaca* strains (HY13M, TE9M, BY2T, TE9T, and CH6T) were collected from different regions in Xinjiang and subjected to parasitic ability tests (Table 1). The hosts (*C. lanatus*, *C. moschata*, and *L. siceraria*) were sown in pots (diameter = 12 cm, height = 18 cm) containing culture substrate (soil/vermiculite/sand = 2:1:1) after inoculating with *P. aegyptiaca*, in which 50 mg of *P. aegyptiaca* seeds was mixed with 1 kg of culture substrate. All plants were grown in a greenhouse (day and night temperatures were 28 °C and 20 °C, respectively; relative air humidity: 40%; illumination duration: 14 h/d), and 10 replicates were set up for each inoculation.

*Nicotiana benthamiana*, used as a host plant, was grown in a greenhouse (day and night temperatures were 28 °C and 20 °C, respectively; relative air humidity: 40%; illumination duration: 14 h/d). The hosts were germinated in 2.5 L pots filled with culture substrate. The grown seedlings were then transferred to 300 mL pots.

### 2.2. Determination of the Parasitic Ability of Different P. aegyptiaca Populations

To determine the parasitic ability of different *P. aegyptiaca* populations, the seeds of *P. aegyptiaca* (HY13M, TE9M, BY2T, TE9T, and CH6T) were used to inoculate *C. lanatus*, *C. moschata*, and *L. siceraria*. Fifty-five days after sowing the host seeds, the substrate adhering to the plant roots was washed off. *P. aegyptiaca* was investigated in different parasitic stages, including the S1 (“haustorium-expanding” stage), S2 (“spider tubercle” stage), S3 (“sprout” stage), and S4 (“underground pre-emergence tissue” stage). The number and fresh weight of *P. aegyptiaca* were recorded.

### 2.3. Transcriptome Analysis of HY13M and TE9M Parasitizing C. lanatus

#### 2.3.1. RNA Extraction, cDNA Library Construction, and Sequencing

Each sample of 100 mg of HY13M or TE9M parasitizing *C. lanatus* at different parasitic stages, namely S1 (D−HY1, D−TE1), S2 (D−HY2, D−TE2), and S3 (D−HY3, D−TE3), were used. The total RNA was isolated from tissue frozen in liquid nitrogen using Trizol Reagent (Invitrogen, Gaithersburg, MD, USA) according to the recommendations of the manufacturer. High-quality RNA, with 28S/18S of more than 1.5 and an absorbance 260/280 ratio between 1.8 and 2.2, was used for library construction and sequencing. The Illumina HiSeq2500 library was constructed according to the manufacturer’s instructions (Illumina, San Diego, CA, USA). mRNA was purified from the total RNA by using magnetic beads with oligo (dT) (Thermo Scientific, Waltham, MA, USA). mRNA was randomly cleaved by adding fragmentation buffer. First-strand cDNA was synthesized using random hexamers. Double-stranded cDNA (dscDNA) was synthesized by adding the buffer, dNTPs, RNase H, and DNA polymerase, and it was then purified using AMPure XP beads. The purified dscDNA was end-repaired, added to an A-tail, and linked with sequencing adapters. AMPureXP beads (Beckman Coulter, CA, USA) were used to choose suitable fragments. The sequencing library was enriched with PCR amplification. The library quality was evaluated on the Agilent Bioanalyzer 2100 system (Agilent, California, USA). The Illumina sequencing platform was Hiseq X ten. The RNA library construction and sequencing were performed at Biomarker Technologies Co. Ltd., Beijing, China. The fragments per kilobase million (FPKM) value of each gene in each sample was calculated using cufflinks [32]. Read counts for each gene were calculated using htseq-count [33]. Differential expression analyses were performed using DESeq2 [34]. The differentially expressed genes (DEGs) with padj < 0.05 and |log_2_fold-change| ≥ 1 were identified. The principal component analysis (PCA) was performed using the ggplot2 package in R (version 4.1.2, ggplot2) software.

#### 2.3.2. Quantitative Real-Time PCR (qRT-PCR)

Nine genes were randomly selected to verify the transcriptome data using qRT-PCR. All primers were designed using Premier Primer 5.0 software (Appendix A) and synthesized by Shanghai ShengGong Biological Engineering Technology Service Co., Ltd. (Shanghai, China). cDNA was synthesized using 1.0 µg of RNA and a First-Strand cDNA Synthesis Kit (SENO Biological Technology Co., Ltd., Zhangjiakou, Hebei, China). The expression levels of the nine genes were determined using an ABI 7500 Fast Real-Time PCR system with SYBR Green chemistry and PaTubulin1 as the housekeeping gene. The reaction was conducted as follows: 95 °C for 3 min, followed by 35 cycles of 95 °C for 5 s, 56 °C for 30 s, and 72 °C for 34 s. Each reaction was performed with three biological replicates, and gene expression levels were calculated using the 2^−ΔΔCt^ method.

### 2.4. Proteome Analysis of HY13M and TE9M Parasitizing C. lanatus

#### 2.4.1. Protein Extraction and Digestion

Protein extraction and digestion were performed based on a previously published protocol [35]. The digested sample was then desalted using a 50 mg tC18 SepPak cartridge (Waters Corporation, Milford, MA, USA), as described previously [36]. For each sample, the tryptic peptides were dissolved in deionized water containing 2% acetonitrile and 0.1% (*v*/*v*) formic acid to a concentration of 500 ng/μL. The pooled peptide sample containing HY13M and TE9M at the S1, S2, and S3 stages (as a QC sample) was analyzed using LC-MS/MS in the data-dependent acquisition (DDA) mode to construct the data-independent acquisition (DIA) spectral library. For retention time calibration, the iRT-standard peptides (Biognosys, Schlieren, Switzerland) were added into the pooled sample and into each sample at a ratio of 1/10 by volume.

#### 2.4.2. Liquid Chromatography–Mass Spectrometry Analysis

A nanoACQUITY UPLC System (Waters Corporation, Milford, MA, USA), which was equipped with a self-packed tunnel-frit [37] analytical column (ID 75 μm × 50 cm length) packed with ReproSil-Pur 120A C18-AQ 1.9 μm (Dr. Maisch GmbH, Ammerbuch-Entringen, Germany) at 40 °C and connected to a Q Exactive HF Hybrid Quadrupole-Orbitrap mass spectrometer (Thermo Scientific, Bellefonte, PA, USA), was used for liquid chromatography–mass spectrometry (LC-MS) analysis. The peptides were separated by a 135 min gradient using mobile phases including Solvent A (0.1% (*v*/*v*) formic acid) and Solvent B (acetonitrile with 0.1% formic acid). With a flow rate of 250 nL/min, the gradient started with a 40 min equilibration maintained at 2% of B and set as the following segments: 2 to 8% of B for 8 min, 8 to 25% of B for 90 min, 25% to 48% of B for 5 min, 48 to 80% of B for another 5 min, followed by 80% of B wash for 10 min and the last equilibrium to 2% B for the last 20 min.

DDA and DIA analyses were conducted according to the previous studies [38,39] using a Q Exactive HF Hybrid Quadrupole-Orbitrap mass spectrometer (ThermoFisher, Shanghai, China).

#### 2.4.3. LC-MS Data Analysis

The DDA data files’ FASTA sequence database was searched with Spectronaut (version 14.4.200727.47784, https://www.biognosys.com/) (Biognosys) software. The database was downloaded from http://www.uniprot.org (accessed on 1 June 2022). The iRT peptide sequence was added (Biognosys|iRT Kit|). The parameters were set as follows: enzyme, trypsin; maximum missed cleavages, 2; fixed modification, carbamidomethyl (C); dynamic modification, oxidation (M) and acetyl (Protein N-term). All reported data were based on 99% confidence for protein identification, as determined by a false discovery rate (FDR = N(decoy) × 2/(N(decoy) + N(target))) of ≤1%. A spectral library was constructed by importing the original raw files, and the DDA results were searched using Spectronaut Pulsar X TM_12.0.20491.4 (Biognosys).

The DIA data were analyzed with SpectronautTM 14.4.200727.47784 and used to search the previously constructed spectral library. The main software parameters were set as follows: retention time prediction type, dynamic iRT; interference on MS2 level correction, enabled; and cross-run normalization, enabled. All results were filtered based on a Q value cutoff of 0.01 (equivalent to FDR < 1%).

#### 2.4.4. Quantitative Data Analysis

To test for significance, Student’s *t*-tests were performed. Any protein with a *p*-value of less than 0.05 and log_2_ fold-change higher than 1.5 or lower than −1.5 was defined as a differentially expressed protein (DEP).

Functional analyses of the DEGs and DEPs in terms of Gene Ontology (GO) enrichment (http://geneontology.org/, accessed on 12 January 2022) and Kyoto Encyclopedia of Genes and Genomes (KEGG) enrichment were conducted using the GOseq R package [40] and KOBAS software (2.0) [41], respectively.

#### 2.4.5. Weighted Gene Co-Expression Network Analysis (WGCNA)

The WGCNA package in R (Version 1.69) was used to identify distinct protein modules among all the identified proteins [42,43]. Using the correlation between module eigengenes (MEs) and different samples [44], the modules of interest were identified. Correlations between module membership (MM) and gene significance (GS) were calculated to identify modules of interest.

### 2.5. Verification of the Effect of the Pectinesterase Gene on Parasitism by Means of HIGS

*TR138070_c0_g1* of *P. aegyptiaca*, which encodes a pectinesterase-like protein, was silenced by means of host-induced gene silencing (HIGS). *N. benthamiana* phytoene desaturase (*NtPDS*) was selected as an indicator. Silenced regions were predicted using the SNG-VIGS website (https://vigs.solgenomics.net/) (accessed on 1 June 2022). The predicted region was amplified using forward and reverse oligos flanked with a homologous arm and cloned into a TRV2 vector using BamHI and XhoI. The forward and reverse primer sequences were as follows: *TR138070_c0_g1*-F, gtgagtaaggttaccgaattcCCGGTAAGTACACGGAGAATGTG; *TR138070_c0_g1*-R, cgtgagctcggtaccggatccTGGAGTGGACATAGAGGGTGTCC. The recombinant clones (TRV2: TR138070) containing the insert were confirmed with diagnostic PCR and Sanger DNA sequencing analyses. The correct clone and TRV2 empty vector (TRV2:00) were transferred to *Agrobacterium* strain GV3101. Single colonies were used to inoculate a broth culture (5 mL), followed by a secondary broth culture (50 mL) in the presence of rifampicin and kanamycin antibiotics. The colonies were then grown overnight at 28 °C. The next day, 50 mL of the cell culture was pelleted by centrifugation at 3000 rpm for 10 min. The recovered pellet was dissolved in infiltration buffer (10 mM 2–N-morpholino ethanesulfonic acid; 10 mM MgCl2; 250 μM acetosyringone in double-deionized water), adjusted to an optical density (OD) of 1.0 (600 nm), and then incubated at room temperature for 3 h. Immediately before infiltration, a culture of pTRV1 and pTRV2 in a 1:1 (*v*/*v*) ratio was mixed into the infiltration buffer (0.0976 g of MES in 100 mL of water (5 mM), adjusting a pH of 5.6). The *Agrobacterium* mixture was injected into *N. benthamiana* leaves grown for 30 days by agrobacterium infiltration with a 2.0 mL syringe. A booster dose was given 1 week later.

After 12 days of agroinfiltration, the host plant was transferred into a 4 L pot containing natural and vermiculite culture soil mixed with *P. aegyptiaca* seeds (50 mg/kg soil). The total weight and number of *P. aegyptiaca* were determined in the different parasitic states (S1, S2, S3, and S4). The total RNA was extracted from *P. aegyptiaca* for the analysis of target gene expression. The data were analyzed using Student’s *t*-tests. Differences with a *p*-value less than 0.05 were considered statistically significant.

## 3. Results

### 3.1. Parasitic Ability of Different P. aegyptiaca Populations

Five *P. aegyptiaca* populations were tested for their parasitic ability on different hosts, including *C. lanatus*, *L. siceraria*, and *C. moschata*. The *P. aegyptiaca* number at different stages (S1, S2, S3, and S4) (Figure 1A), total number (Figure 1B), and fresh weight (Figure 1C) of HY13M were reduced compared to those of TE9M, CH6T, BY2T, and TE9T on *C. lanatus* hosts; the total number and fresh weight of HY13M was reduced by 87.93% and 90.87%, respectively, compared with TE9M (Figure 1B,C). Additionally, the *P. aegyptiaca* number at different stages (S1, S2, S3, and S4) (Figure 1E), total *P. aegyptiaca* number (Figure 1F), and fresh weight (Figure 1G) of HY13M were reduced compared to those of TE9M, CH6T, BY2T, and TE9T when on *L. siceraria* hosts; the total number and fresh weight of HY13M was reduced by 79.78% and 98.51%, respectively, compared with TE9M (Figure 1F,G). The results of the parasitic ability test on *C. moschata* showed that the number of HY13M significantly decreased in the S1, S2, and S3 stages, but not in the S4 stage, compared to TE9M (Figure 1I); the total number of HY13M decreased by 47.18% compared with TE9M (Figure 1J), but the fresh weight of HY13M was not significantly different from that of TE9M (Figure 1K).

In conclusion, HY13M exhibited significantly decreased parasite number and fresh weight compared with TE9M when parasitizing *C. lanatus* and *L. siceraria* (Figure 1D,H). Therefore, HY13M and TE9M had relatively weak and strong parasitic abilities, respectively, for *C. lanatus*, and were selected for the subsequent parasitism study.

### 3.2. Transcriptomic Analysis and Function Analysis of DEGs

To analyze the parasitism mechanism of *P. aegyptiaca*, RNA sequencing (RNA-Seq) was performed to evaluate different populations of *P. aegyptiaca* (HY13M and TE9M) parasitizing *C. lanatus* at different parasitic stages. The PCA analysis results showed that the three replicates of each *P. aegyptiaca* population clustered together (Figure 2A). The expression trend of selected genes was consistent with their expression trend in the transcriptome analysis by means of qRT-PCR, showing the reliability of the RNA-Seq expression data (Appendix A). In the three parasitic stages, 23,902, 30,288, and 25,459 genes were significantly upregulated in D−TE1, D−TE2, and D−TE3, respectively, and 23,090, 33,611, and 20,474 genes were significantly downregulated in D−TE1, D−TE2, and D−TE3, respectively (Figure 2B). Among the DEGs of the “D−TE3 vs. D−TE2” comparison group, the most significantly enriched terms were “disaccharide metabolic”, “oligosaccharide metabolic”, “cell wall organization or biogenesis”, “external encapsulating structure organization”, and “cell wall organization” in the BP category; “phosphoenolpyruvate carboxykinase activity”, “carboxy-lyase activity”, and “carbon–carbon lyase activity” in the MF category; and “ribosome”, “chloroplast”, “cell wall”, and “external encapsulating structure” in the CC category (Figure 2C).

In addition, the results of the KEGG enrichment analysis showed that among the DEGs from the “T-HY1 vs. T-TE1” comparison, the most enriched pathways were metabolism-related pathways; among them, “pentose and glucuronate interconversions”, “starch and sucrose metabolism”, “fructose and mannose metabolism”, “glycolysis/gluconeogenesis”, and “two-component system”, relating to cell wall metabolism, were significantly enriched (Figure 2D). Meanwhile, the upregulated DEGs in the D−TE2 vs. D−TE1 comparison and upregulated DEGs in the D−TE3 vs. D−TE2 comparison contained cell wall metabolism genes, such as glucan endo-1,3-beta-D-glucosidase, beta-glucosidase, acidic endochitinase, expansin-like, polygalacturonase-1 non-catalytic subunit beta-like, hydrolase, pectinesterase-like, endoglucanase 17-like protein, etc. Therefore, cell wall metabolism is closely related to *P. aegyptiaca* parasitism.

### 3.3. Proteomic Analysis of DEPs and Function Analysis

The protein quantification showed that there were 17,975 peptides and 6564 proteins in the 18 samples. The PCA showed that the samples from the same treatment were clustered together and that the parasitic samples of different *P. aegyptiaca* populations were scattered (Figure 3A). Screening for differentially expressed proteins (DEPs) resulted in 499 DEPs from the “D−TE1 vs. D−HY1” comparison (Figure 3B). The GO enrichment analysis showed that among the DEPs from the “D−TE1 vs. D−HY1” comparison, the most significantly enriched terms were “cell wall organization”, “external encapsulating structure organization”, and “cell wall organization or biogenesis” in the BP category; “glycerophosphodiester phosphodiesterase activity”, “hydrogen-translocating pyrophosphatase activity”, and “nutrient reservoir activity” in the MF category; and “cell wall”, “external encapsulating structure”, and “extracellular region” in the CC category (Figure 3C). Moreover, the KEGG enrichment analysis showed that among the DEPs from the “D−TE1 vs. D−HY1” comparison, the “Two-component system” pathway, which is associated with cell wall metabolism, was significantly enriched (Figure 3D). In conclusion, the DEPs related to cell wall activity are highly involved in *P. aegyptiaca* parasitism.

To analyze the parasitism mechanism of *P. aegyptiaca* further, a module which contained 1936 proteins was identified according to the heatmap of module–trait correlations from the WGCNA (Figure 3E, module shown in turquoise). This module was positively correlated with D−TE3 (r = 0.7 *p* = 0.001) (Figure 3E). Within the turquoise module, cell-wall-degrading enzymes were identified, including glucan endo-1,3-beta-glucosidase, chitinase-like, pectinesterase-like, beta-galactosidase, beta-galactosidase 5-like, xylosyltransferase, acidic endochitinase, endoglucanase 17-like, alpha-xylosidase, beta-fructofuranosidase, and fasciclin-like arabinogalactan proteins. Cell-wall-modifying enzymes were identified in the turquoise module, such as xylose isomerase and extensin-like protein. Expansin-like protein was also detected in the turquoise module. This research demonstrated that proteins related to cell wall activity are highly correlated with TE9M parasitism.

### 3.4. Combined Transcriptome and Proteome Analysis Revealed the Involvement of Cell Wall Metabolism Enzymes in P. aegyptiaca Parasitism

To find evidence that *P. aegyptiaca*’s parasitic ability is affected by both transcription and protein levels, the enriched KEGG pathways of both the DEGs and DEPs between HY13M and TE9M when parasitizing *C. lanatus* were analyzed together. In the comparison of the primary stages (D−HY1 vs. D−TE1), the “two-component system” pathway was enriched in both the DEGs and DEPs (*p*-value (DEGs) = 0.038; *p*-value (DEPs) = 0.019), suggesting that this pathway may play a key role in the different parasitic abilities of HY13M and TE9M. Among all the DEGs and DEPs involved, those involved in the two-component pathway were *TR13022_c0_g1* and *TR13022_c0_g2*, which encode pectinesterases (Figure 4). The KEGG diagram showed that these two proteins were in a class of virulent autoregulation genes as plant cell-wall-degrading enzymes (Figure 4).

### 3.5. HIGS Confirmed Pectinesterase-like Gene Involved in P. aegyptiaca Parasitism

*P. aegyptiaca* pectinesterase-like gene (*TR138070_c0_g1*) was upregulated in D−TE1 compared to D−HY1. The phylogenetic analysis for *TR138070_c0_g1*, *TR13022_c0_g1*, and *TR13022_c0_g2* showed that *TR138070_c0_g1* is closely related to *TR13022_c0_g1,* which was clustered in group I (Figure 5). The gene expression analysis showed that *TR138070_c0_g1* was significantly reduced in *P. aegyptiaca* attached to the TRV2: TR138070-agroinfiltrated host compared with the TRV2:00-agroinfiltrated host at different parasitic stages (S1, S2, S3, and S4) (Figure 6A). The number and fresh weight of *P. aegyptiaca* in the different parasitic stages were significantly reduced after *TR138070_c0_g1* was silenced (S1, S2, S3, and S4) (Figure 6B, 6C, 6D). Compared to the control treatment (TRV2:00), the number of *P. aegyptiaca* decreased by 43.75, 41.17, 38.46, and 80% in the S1, S2, S3, and S4 stages, respectively, after *TR138070_c0_g1* was silenced (Figure 6B). Compared with the TRV2:00-agroinfiltrated treatment, the fresh weight of *P. aegyptiaca* decreased by 60.48% after *TR138070_c0_g1* was silenced (Figure 6C). HIGS of *TR138070_c0_g1* in *P. aegyptiaca* showed that *TR138070_c0_g1* significantly affected the parasitic ability of *P. aegyptiaca*.

## 4. Discussion

Pectinesterases in pathogens might assist their invasion into host tissues [45]. Prior research suggests that fungal pathogens could induce higher activities of cell wall polysaccharide-disassembling enzymes, such as pectinesterase, leading to disease occurrence [46]. Additionally, the virulent pathogen *Rhizopus oryzae* increases the activity of pectinesterase, causing pumpkin fruit rot [47]. In parasitic plants, *O. cumana* produces pectinesterases in susceptible interactions [28]. In this study, the KEGG and GO analyses showed that the “two-component system” pathway was enriched in DEGs and DEPs, of which two pectinesterases were detected in both the proteome and transcriptome. Further verification demonstrated that the *TR138070_c0_g1* gene of *P. aegyptiaca* encoding pectinesterase affected its parasitic ability, as confirmed by means of HIGS. Thus, pectinesterases may contribute to the degradation of the host cell wall, facilitating the penetration of the parasite’s haustorium into the host root.

The composition of the cell wall is a major factor that induces immune responses during the pathogenetic process [48]. For example, pectinesterase can also promote pectin degradation and suppress immunity, apart from the direct impact of pectinesterases on the cell wall structure [49]. Moreover, pectin interacts with pattern recognition receptors (PRRs) to exert immune effects [50]. Thus, pectinesterase may indirectly influence *P. aegyptiaca*’s parasitism via pectin. This study revealed that pectinesterases participate in the parasitism process, possibly through influencing the immune response of *P. aegyptiaca*.

The enzymes of pathogens not only facilitate their invasion, but they also manipulate host cellular processes [51,52]. Similarly, expansin protein can unlock the network of wall polysaccharides, loosening the plant cell wall [53]. Research has also shown that expansin-like proteins can modulate the immune responses in *N. benthamiana* against *Pratylenchus penetrans*, a parasitic nematode [54]. Thus, *P. aegyptiaca* may produce proteins targeting the cell wall to help its invasion, which therefore serve as parasitic factors.

Cell wall metabolism enzymes that were highly correlated with *P. aegyptiaca* parasitism are involved in cell wall degradation and modification. In this study, a large number of proteins related to cell wall metabolism were differentially expressed during *P. aegyptiaca* parasitism based on the transcriptome and proteome analyses, suggesting that cell wall metabolism enzymes play an important role in parasitism; these enzymes include glucan endo-1,3-beta-D-glucosidase, beta-glucosidase, acidic endochitinase, xylose isomerase, extensin-like protein, etc. There have been reports that xylosyltransferases participate in the biosynthesis of xyloglucan, which is an abundant hemicellulosic component of the cell wall [55]. Moreover, extensin-like protein is responsible for cell wall modification [56]. On the other hand, the host cell wall undergoes alterations when parasitized by parasites, such as partial wall dissolution and shredding [57,58], which aids the parasitic plant in invading the host. During the process of Orobanchaceae plants penetrating the host root, the activities of cell-wall-degrading enzymes, such as pectolytic enzymes, cellulase, and polygalacturonase, become evident and are present in infecting tissues [59]. Cell-wall-modifying enzymes are also significantly upregulated at this stage [60]. For example, in Orobanchaceae parasitic species, genes that encode cell-wall-modifying enzymes are highly upregulated during haustorial development [22]. Research has found that β-1,4-glucanase plays an important role in parasitism for the Orobanchaceae plant *Phtheirospermum japonicum* [60]. In the infective stages of dodder, the expression of genes encoding cell-wall-modifying enzymes, such as pectin lyase, pectin methyl esterase, and expansins, is enhanced [61]. Additionally, the transcriptome investigation found that cell-wall-modifying enzymes of *T. versicolor* were upregulated at the host–parasite interface [30]. Thus, *P. aegyptiaca* can produce cell-wall-degrading proteases and cell-wall-modifying proteases to penetrate host roots.

The same proteins are associated with cell wall metabolism, including beta-glucosidase, acidic endochitinase, and endoglucanase 17-like, and pectinesterase proteins were detected at both the transcript and protein levels in this study. In summary, cell wall proteases were highly correlated with the parasitism of *P. aegyptiaca* and might play an important role in parasitic interactions at both the transcript and protein levels.

## 5. Conclusions

In summary, the metabolism pathways of the cell wall play important roles in the differentiation of the parasitic capacity of *P. aegyptiaca*. Enzymes involved in cell wall metabolism affect parasitic capacity at the transcript and protein levels, with pectinesterase having a significant impact on parasitic capacity.

## Figures and Tables

**Figure 1 plants-13-00869-f001:**
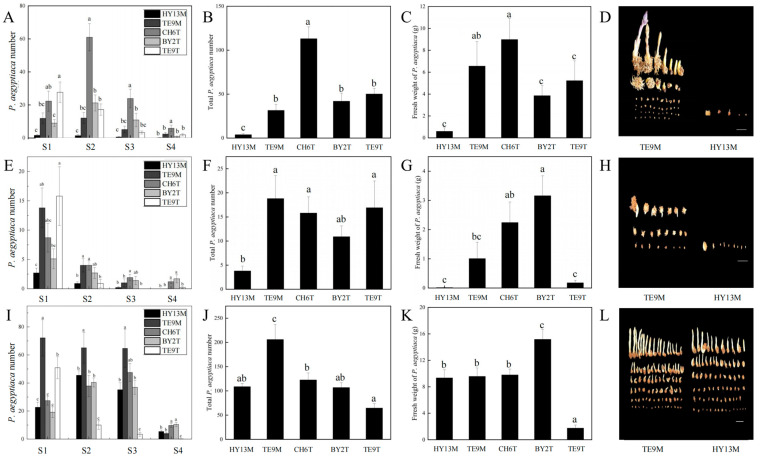
The parasite number of different *P. aegyptiaca* populations parasitizing *C. lanatus*, *L. siceraria*, and *C. moschata*. (**A**) The number of *P. aegyptiaca* parasitizing *C. lanatus* at different parasitic stages. (**B**) The total number of *P. aegyptiaca* parasitizing *C. lanatus*. (**C**) Fresh weight of *P. aegyptiaca* parasitizing *C. lanatus*. (**D**) Presentation of TE9M and HY13M parasitizing *C. lanatus*. (**E**) The number of *P. aegyptiaca* parasitizing *L. siceraria* at different parasitic stages. (**F**) The total number of *P. aegyptiaca* parasitizing *L. siceraria*. (**G**) Fresh weight of *P. aegyptiaca* parasitizing *L. siceraria*. (**H**) Presentation of TE9M and HY13M parasitizing *L. siceraria*. (**I**) The number of *P. aegyptiaca* parasitizing *C. moschata* at different parasitic stages. (**J**) The total number of *P. aegyptiaca* parasitizing *C. moschata*. (**K**) Fresh weight of *P. aegyptiaca* parasitizing *C. moschata*. (**L**) Presentation of TE9M and HY13M parasitizing *C. moschata*. Data are presented as mean ± SE. The letters above each bar indicate significant (*p* ≤ 0.05) differences between groups based on Duncan’s test. Scale bars = 2 cm.

**Figure 2 plants-13-00869-f002:**
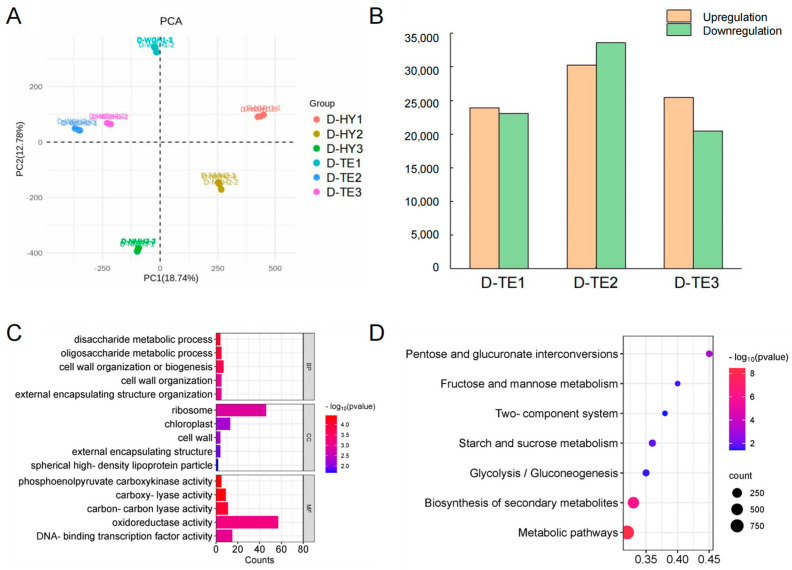
Overall transcriptome analysis of the HY13M and TE9M parasitism groups. (**A**) Principal component analysis (PCA) of transcriptome data of HY13M and TE9M parasitizing *C. lanatus* at S1, S2, and S3 stages. (**B**) The number of upregulation and downregulation genes in TE9M compared to HY13M at S1, S2, and S3 stages. (**C**) GO enrichment of DEGs in “D−TE3 vs. D−TE2” comparison (*p* < 0.05). (**D**) KEGG enrichment of DEGs in “T−HY1 vs. T−TE1” comparison (*p* < 0.05).

**Figure 3 plants-13-00869-f003:**
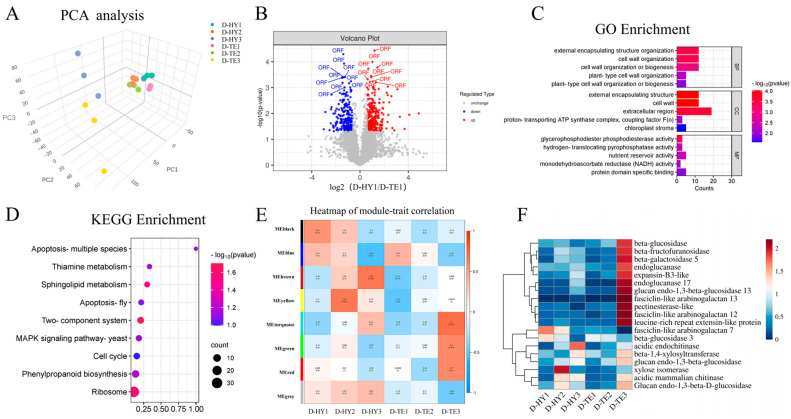
Overview of differential proteomics based on DIA-based liquid chromatography–mass spectrometry analysis. (**A**) Principal component analysis based on protein expression. (**B**) Volcano plots of DEP abundance in “D−HY1 vs. D−TE1” comparison; *t*-test-based significance values (log_10_ (*p*-value)) are plotted versus log_2_ (protein quantity ratio for all proteins between D−HY1 and D−TE1). Upregulated proteins with *p* < 0.05 and log_2_ fold-change > 1.5 are plotted in red; downregulated proteins with *p* < 0.05 and log2 fold-change < −1.5 are plotted in blue. (**C**) GO enrichment analysis of DEPs in “D−HY1 vs. D−TE1” comparison (*p* < 0.05). (**D**) Top 9 terms of KEGG enrichment analysis of DEPs in “D−HY1 vs. D−TE1” comparison. (**E**) Correlation matrix of module eigengene values obtained for stage-specific parasitic traits of two *P. aegyptiaca* populations (HY13M andTE9M) parasitizing *C. lanatus*. The WGCNA grouped proteins into modules based on the patterns of their co-expression. Each of the modules was labeled with a unique color as an identifier. Eight modules were identified; each module eigengene was tested for correlation with stage-specific parasitic traits. (**F**) Clustering heatmap and annotation of cell wall metabolism proteins in turquoise module; deep blue to red indicates the gene expression level from low to high. Expression levels are normalized.

**Figure 4 plants-13-00869-f004:**
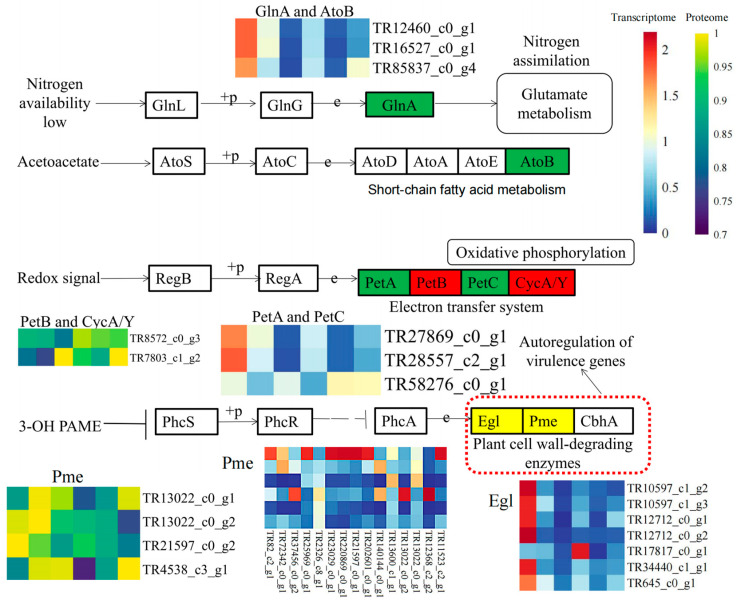
Integrated analysis of the transcriptome and proteome and heat map analysis. Pathway maps and expression heat maps of all DEGs and DEPs in the two-component system. Blue to red indicates the gene expression level from low to high. Deep blue to yellow indicates the protein expression level from low to high. Expression levels are normalized. The red dotted lines show the pathways that are enriched in both the transcriptome and proteome, and the genes Pme *TR13022_c0_g1* and *TR13022_c0_g2* that are enriched in both the transcriptome and proteome.

**Figure 5 plants-13-00869-f005:**
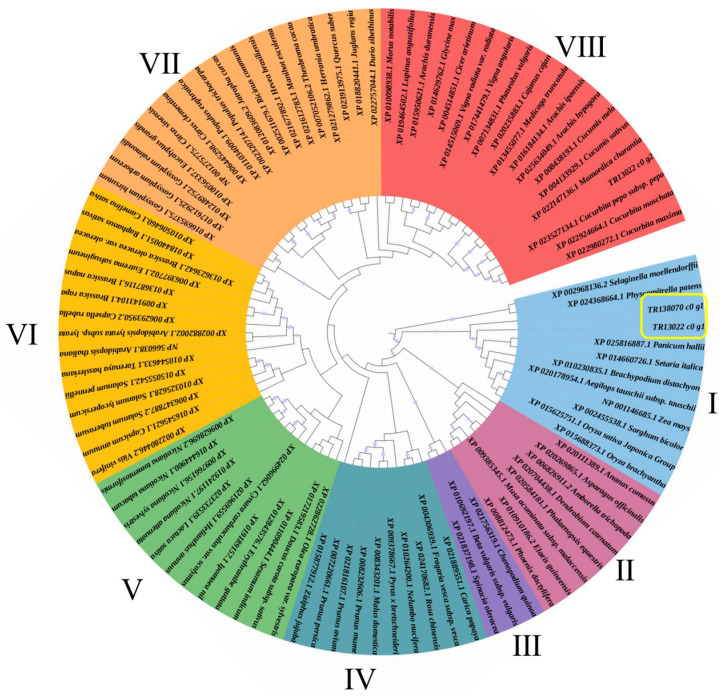
Phylogenetic tree of the *TR138070_c0_g1, TR13022_c0_g1*, and *TR13022_c0_g2* and pectinesterase gene in multiple species. Phylogenetic distance calculated using the MEGA 6 program and the Maximum Likelihood (ML) method using a bootstrap value of 500. Based on the protein sequence similarities between different species, the members were divided into 8 groups (I–VIII). The yellow box is the *TR138070_c0_g1* and *TR13022_c0_g1* genes.

**Figure 6 plants-13-00869-f006:**
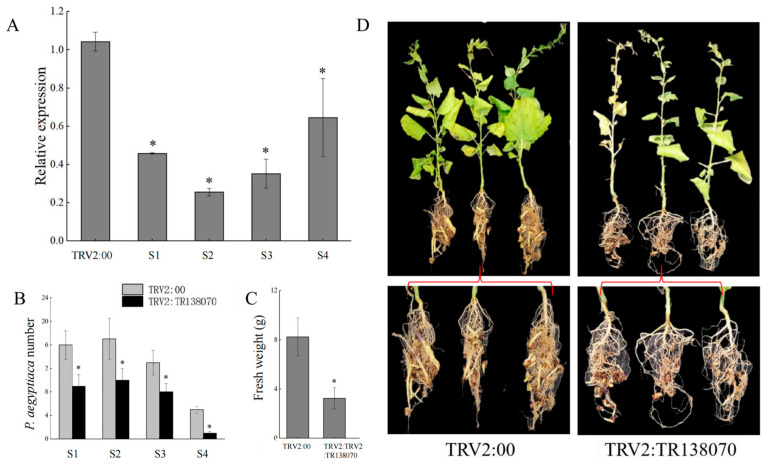
Effect of *TR138070_c0_g1* silencing on *P. aegyptiaca* parasitism. (**A**) Transcript level of *TR138070_c0_g1* in TRV2:TR138070-silenced line using *P. aegyptiaca* tissues at different stages (S1, S2, S3, and S4). The expression level of each gene is displayed after normalization with the internal housekeeping gene *PaTubulin1*. All analyses were performed using three biological replicates. Data are represented as the average ± SE (*n* = 3). (**B**) Number of *P. aegyptiaca* at different stages (S1, S2, S3, and S4) attached to the TRV2:*00* and TRV2:*TR138070* host. (**C**) Fresh weight of *P. aegyptiaca* attached to the TRV2:*00* and TRV2:*TR138070* host. (**D**) Effect of TRV2:TR138070 silencing on the development of *P. aegyptiaca* compared to the vector (TRV2:00) line. Bars represent the average ± SE (*n* = 5) value from two different experiments with five independent host plants. Statistical differences were calculated with Student’s two-tailed *t*-test (*p* < 0.05). The asterisk on the bar indicates a significant difference between the TRV2-mediated silenced line (TRV2:TR138070) compared to the vector control plants (TRV2:00).

**Table 1 plants-13-00869-t001:** Information on the five *P. aegyptiaca* populations used in this study.

Population Name	Collection Time	Collection Location	Host	Longitude, Latitude
TE9M	2017	4th company, 163rd regiment, 9th division, Emin County, Tacheng, Xinjiang	Melon	82°55′10″ E, 46°47′59″ N,
CH6T	2017	3rd company, Junhu farm, 6th division, Hutubi County, Changji Hui Autonomous Prefecture, Xinjiang	Processing tomato	87°0′19″ E, 44°1′27″ N,
BY2T	2017	7th company, 21st regiment, 2nd division, Yanqi County, Bayingolin Mongolian Autonomous Prefecture, Xinjiang	Processing tomato	86°18′54″ E, 42°9′36″ N,
TE9T	2017	4th company, 163rd regiment, 9th division, Emin County, Tacheng, Xinjiang	Processing tomato	82°55′10″ E, 46°47′59″ N,
HY13M	2017	Naomaohu farm, 13th division, Yiwu County Hami, Xinjiang	Melon	94°58′56″ E, 43°55′52″ N,

## Data Availability

The data used to support the findings of this study are available from the corresponding author upon request.

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
