# Peer review of "Integrated Transcriptome and Proteome Analysis Reveals That Cell Wall Activity Affects Phelipanche aegyptiaca Parasitism"

_plants, 2024, doi:10.3390/plants13060869_

Round 1

Reviewer 1 Report

Comments and Suggestions for Authors The paper by Chen et al. reports an investigation of the virulence of different P. aegyptiaca populations towards potential hosts. The work proceeds with transcriptomic analyzes conducted during the interaction between 2 differently virulent populations of the parasitic plant towards the host Cucurbita lanatus. Subsequently, it continues with a GO enrichment analysis at different stages of the infection and proceeds with an analysis of the secretome focused mainly on aspects correlated with wall alterations. I believe that the multiple objectives pursued by the authors are not explained organically. The study contains a lot of information but it is unclear which alterations concern the host plant and which concern the parasitic plant. Reading the text is difficult and even the bibliographical references in the introduction and discussion are not presented in an organic way. In the discussion it is difficult to understand what the progress of this research is compared to what is already known in the literature.      

Reviewer 2 Report

Comments and Suggestions for Authors

Dear authors,

This is a research work that was well structured. I will like to thank the authors for a job well done. The manuscript investigates the parasitic behaviour of two populations of Phelipanche aegyptiaca, HY13M and TE9M, with a focus on their interaction with Citrullus lanatus. Transcriptome and proteome analyses were used to explore the molecular basis of the observed differences in parasitism. This findings highlight the involvement of the cell wall metabolism pathway, particularly the role of pectinesterases encoded by the TR138070_c0_g1 gene, in influencing the parasitic ability of P. aegyptiaca populations. It is obvious that the study contributes valuable insights into the mechanisms underlying parasitism in P. aegyptiaca.

In view of improving the write up for publication, I will propose some suggestions and comments:

1.     Review the references, ensuring consistent formatting, particularly regarding single-authored articles such as references 3 and 68. Verify that author names are presented uniformly throughout.

2.     It is important to elucidate the rationale behind selecting Citrullus lanatus, Cucurbita moschata, and Lagenaria siceraria. This information could be highlighted as a key focal point in Table 1.

3.     Lines 95-97 seems to represent same meaning as in line 86 and 87. Consider revising to avoid repetition.

4.     Lines 144-150:  Consider treating lines 144-150 as an independent section, specifically focusing on the transcriptomes analysis. If my understanding is accurate, this segment should be delineated as a distinct section dedicated to transcriptomes.

5.     Check out the line spacing between lines 174 and 175.

6.     Line 196: delete “validation”

7.     Lines 271 to 277: The title of figures and tables. Ensuring a chronological sequence and maintaining consistency in the lettering of the figure will enhance clarity and reduce potential confusion in the legend.

8.     The figures and tables are well designed and effectively support the results presented in the text. Ensure that the figures are labeled appropriately, and legends provide sufficient information for interpretation.

9.   The implications of the findings for developing prevention and control methods in broomrape are discussed. Consider expanding on potential PRACTICAL APPLICATIONS and avenues for future research.

10.  The overall language is clear and concise, contributing to the accessibility of the manuscript. Carefully proofread the manuscript for grammatical and typographical errors.

11. The abstract provides a comprehensive overview of the study, but the flow could be improved for better coherence. I recommend that the authors refine the structure to enhance the seamless connection between the experimental setup, results, and conclusions.

I believe that addressing these points will significantly improve the manuscript and contribute to its successful publication. 

Thank you for your contribution to the field.

Best regards,

Comments on the Quality of English Language

1. The overall language is clear and concise, contributing to the accessibility of the manuscript. Carefully proofread the manuscript for grammatical and typographical errors.

Reviewer 3 Report

Comments and Suggestions for Authors

The idea of this research is of importance and the write up of the article is good. There are a few comments, which point out the deficiencies in the manuscript and suggest for its improvement.

The text has minor language mistakes. Some of these are highlighted in the attached file. The authors may have a carefull recheck.

L 89-90. Research gap is missing, and it should be added here.

L 104. For the plant growing conditions, the day temperature was kept 20 C, and night temperature as 28 C. Actually this should be vice versa. Later on in the text (L 192), a single and different temperature (24 C) has been given for Nicotiana benthamiana, and the day night temerature differences are missing there. Further, the day length is also different for the two cases. This is required to be rechecked and corrected if needed.

Write up (particulary flow) of the materials and methods section is required to be improved. The authors may recheck their materials and methods in the perspective of this classic statement "Methods should be detailed and transparent, enabling other researchers to replicate the study's procedures and obtain similar results."

 Details about handling of data and statistical analysis are missing.

A separate conclusions section is missing there.

Comments on the Quality of English Language

The idea of this research is of importance and the write up of the article is good. There are a few comments, which point out the deficiencies in the manuscript and suggest for its improvement.

The text has minor language mistakes. Some of these are highlighted in the attached file. The authors may have a carefull recheck.

L 89-90. Research gap is missing, and it should be added here.

L 104. For the plant growing conditions, the day temperature was kept 20 C, and night temperature as 28 C. Actually this should be vice versa. Later on in the text (L 192), a single and different temperature (24 C) has been given for Nicotiana benthamiana, and the day night temerature differences are missing there. Further, the day length is also different for the two cases. This is required to be rechecked and corrected if needed.

Write up (particulary flow) of the materials and methods section is required to be improved. The authors may recheck their materials and methods in the perspective of this classic statement "Methods should be detailed and transparent, enabling other researchers to replicate the study's procedures and obtain similar results."

 Details about handling of data and statistical analysis are missing.

A separate conclusions section is missing there.

Author Response

Dear Reviewer 3
